# Nutrition-Related Knowledge, Diet Quality, Lifestyle, and Body Composition of 7–12-Years-Old Polish Students: Study Protocol of National Educational Project Junior-Edu-Żywienie (JEŻ)

**DOI:** 10.3390/nu16010004

**Published:** 2023-12-19

**Authors:** Jadwiga Hamulka, Ewa Czarniecka-Skubina, Krystyna Gutkowska, Małgorzata Ewa Drywień, Marta Jeruszka-Bielak

**Affiliations:** 1Department of Human Nutrition, Institute of Human Nutrition Sciences, Warsaw University of Life Sciences (SGGW-WULS), 166 Nowoursynowska St., 02-787 Warsaw, Poland; jadwiga_hamulka@sggw.edu.pl (J.H.); malgorzata_drywien@sggw.edu.pl (M.E.D.); 2Department of Food Gastronomy and Food Hygiene, Institute of Human Nutrition Sciences, Warsaw University of Life Sciences (SGGW-WULS), 166 Nowoursynowska St., 02-787 Warsaw, Poland; ewa_czarniecka-skubina@sggw.edu.pl; 3Department of Food Market and Consumer Research, Institute of Human Nutrition Sciences, Warsaw University of Life Sciences (SGGW-WULS), 166 Nowoursynowska St., 02-787 Warsaw, Poland; krystyna_gutkowska@sggw.edu.pl

**Keywords:** pupils, dietary habits, nutrition-related knowledge, nutritional status, body composition, parents, teachers, education program

## Abstract

Increasing students’ nutrition knowledge is an important goal of school education which may improve their lifestyle and consequently reduce the incidence of non-communicable chronic diseases, including obesity. This research aimed at assessing the dietary habits, nutrition-related knowledge and attitudes, and nutritional status of Polish students aged 7–12 years. Additional objectives included recognizing nutrition-related knowledge among their parents and teachers as well as the school food and nutrition environment. The final goal was to develop a nutrition education program. This study protocol presents a general and detailed approach for realizing the above-mentioned issues, including conducting quantitative and qualitative research. The study was designed as cross-sectional, covering primary school students from all over Poland, including rural, town, and metropolitan areas. Data on eating habits, lifestyle, nutrition-related knowledge, and attitudes were collected with questionnaires. Measurements of body weight, height, waist and hip circumferences, and handgrip strength were performed in accordance with the International Standards for Anthropometric Assessment (ISAK) recommendations. Body composition was assessed with bioelectrical impedance analysis. Ultimately, 2218 schools from all 16 voivodeships in Poland registered for the project. In total, quantitative data were collected among 27,295 students, 17,070 parents, and 2616 teachers. Anthropometric measurements were taken among 18,521 students. The results allow us to develop a multifaceted educational program based on knowledge and adapted to the perception of students. Our research may contribute to the identification of subpopulations of children and adolescents at risk of excessive body weight and define the predictors of obesity risk in Poland.

## 1. Introduction

Adequate nutrition of children and adolescents is one of the key determinants in their proper physical and mental development. Moreover, eating habits established in childhood and adolescence tend to persist and may be difficult to modify in later stages of life, which is often neglected by adults [1,2]. Incorrect eating behaviours, such as irregular meals; skipping breakfast at home and at school; eating larger portions of meals; snacking between meals; increased consumption of sugar-sweetened beverages, sweets, salty snacks, fast food, and generally ultra-processed foods, while reduced consumption of vegetables and fruit, whole grain products, dairy products, and fish are frequently observed among children and adolescents in developed countries [3,4], also in Poland [5,6,7], in both urban and rural environments [8,9]. They, in turn, result in an unbalanced diet, which on the one hand has excessive energy content, specifically derived from high-energy foods, fats, and high-glycaemic-index carbohydrates, and on the other hand, is deficient in micronutrients, dietary fibre, and biologically active compounds [10,11,12]. An unhealthy diet is often associated with a lack of physical activity and a sedentary lifestyle. The coexistence of these lifestyle issues promotes weight gain and reduces the exercise capacity of the circulatory system. Additionally, psychological factors such as anxiety and depression may contribute to or result from eating disturbances [13,14].

The above-mentioned abnormalities are direct causes of obesity and, consequently, other diet-related diseases like type 2 diabetes, hypertension, and atherosclerosis, which are already considered common childhood diseases [15,16]. As the prevalence of obesity steadily increases and considering that it originates from multiple factors and is difficult to eliminate, “this pandemic of the 21st century” has become the most serious challenge to public health [17,18] in the context of psycho-social and economic effects [12,19,20,21,22]. According to World Obesity Federation prognosis, the number of obese children in Poland will steadily increase by 4.8% every 5 years in the period 2020–2035, which is defined as a very high rate [23].

It is also worth noting that many eating disorders such as anorexia, bulimia, binge eating disorders, and orthorexia appear in early adolescence [24,25]. Cultural factors, addictions (e.g., cyber addiction), distortion of body image, and unhealthy weight-loss diets widely promoted by the media, but also lifestyle changes including stress and diet influence the incidence of eating disorders [26]. In developed countries, including Poland, eating disorders are a serious and growing social and health problem. Girls and young women are particularly vulnerable to eating disorders, but they can also occur in men and boys, especially during puberty [24,27]. 

Inadequacies in diet and physical activity, and their health consequences, occur with varying intensity depending on age, gender, nutritional knowledge of both adolescents and their parents, susceptibility to dietary trends, family characteristics, parenting style and parents’ lifestyle, parents’ work, and the level of social and living conditions, as well as environmental factors such as school food policy [9,14,28,29].

Parents are the first and key factor in their child’s socialization and shaping their child’s behaviour. The home environment can establish the child’s behaviour to be towards behaviour typical for a healthier lifestyle and thus reduce the risk of eating disorders and body weight abnormalities. Parental attitudes and their own behaviours concerning physical activity [30], healthy eating patterns, screen time, and sleeping [31] are particularly important.

School is the second environment, coming next after family, that influences a proper lifestyle, including the nutrition of children and adolescents. The school’s activities cover formal nutritional education during lessons in accordance with the core curriculum, the meals and foods available at the school (in the canteen, school shop, vending machine), additional activities related to the promotion of a healthy lifestyle, e.g., external and internal educational programs, events organized at the school, e.g., child’s day, days thematically related to nutrition, as well as the frequency of physical education classes and additional physical activities [29,32,33].

Taking into account the presented argumentation, there is an urgent need to integrate both environments, namely school and family, in improving eating habits and behaviours among children and adolescents. Learning about the nutrition-related problems and needs perceived by school actors and families is the first stage in the development of a nutrition education program that may meet the expectations and requirements. Combining both settings will bring advantages in many areas like children and family health, building social bonds, and improving the condition of the natural environment by promoting the ideas of sustainable diets as well reducing food waste, which will in turn bring measurable socio-economic benefits, i.e., improved quality of life and a cost reduction in the treatment of diet-related diseases, as well a as reduced carbon footprint. 

So far, in Poland there have been no nationwide studies, combining quantitative and qualitative approaches, conducted in large groups of primary school-age children and their parents, as well as their teachers, and covering a broad range of nutrition-related issues. This becomes particularly important after the COVID-19 pandemic, which changed lifestyle, including dietary, behaviour, also among children and adolescents [34].

Thus, the objectives of the Junior-Edu-Żywienie (JEŻ) Project were (1) to assess the dietary habits, nutrition-related knowledge and attitudes, and nutritional status, including body composition, of Polish students aged 7–12 years; (2) to evaluate the nutrition-related knowledge and attitudes of their parents as well as (3) among primary school teachers; (4) to evaluate the school food and nutrition environment, and finally (5) to develop the nutrition education program responding to identified needs and problems.

## 2. Materials and Methods

### 2.1. Ethics Approval

The study protocol was approved by the Ethics Committee of the Institute of Human Nutrition Sciences, Warsaw University of Life Sciences, Poland (Resolution No. 18/2022). In addition, the guidelines of the Declaration of Helsinki were followed in the course of the study. The study was explained to the participants before the start (verbally and by leaflets), to the parents/caregivers and teachers during the planned school meetings, and to the pupils during their lessons. Parental/caregiver’s written informed consent was obtained for the participation of their children. The consent form was distributed among parents/caregivers during the school meetings. Children who did not participate in the research were under the care of teachers and attended other school activities during the data collection in their school.

### 2.2. Study Design and Participants

Within the Junior-Edu-Żywienie (JEŻ) Project, a cross-sectional, and observational study among school children aged 7–12 years, their parents, and teachers was carried out in the period from April 2022 to November 2023. It consisted of following main stages: recruitment of schools and participants (April 2022–January 2023), the pilot study (April–May 2022), and the final fieldwork (June 2022–November 2023). The overview of stages and activities undertaken during the Junior-Edu-Żywienie (JEŻ) Project is presented in Figure 1.

The main idea of the Project was to create a representative sample including 1st–6th grade schoolchildren (aged 7–12 years) from schools located in all 16 voivodeships across Poland. The school class was the smallest sampling unit. This approach resulted from the theory that the students underwent the same school education and were at a similar stage of development. The schools were selected based on the distribution of schools into regions, considering three sizes of the locality (i.e., villages, towns with up to 100,000 inhabitants, and large urban agglomerations). Taking the above into account, five separate macroregions were included: Central (Masovian Voivodeship, Łódź Voivodeship); North-Eastern (Warmian-Masurian Voivodeship, Podlaskie Voivodeship, Lublin Voivodeship); North-West (Pomeranian Voivodeship, West Pomeranian Voivodeship, Kuyavian-Pomeranian Voivodeship and Greater Poland Voivodeship); South-Western (Lubusz Voivodeship, Lower Silesia Voivodeship, Opole Voivodeship and Silesia Voivodeship); South-Eastern (Świętokrzyskie, Lesser Poland and Podkarpackie) (Figure 2), corresponding to the distribution of schools and children aged 7–12.

Recruitment of schools was based on data obtained from the Ministry of Education and Science. The number of primary schools was verified, and an appropriate database was created in 16 voivodeships. Some schools were eliminated because they were dedicated to adults, or ceased to exist, or were branches of other schools, so they were paired and classified as one school. The email invitation to participate in the project was sent to all qualified schools (*n* = 14,698), both public and non-public. Ultimately, 2218 schools (15.1%) from all voivodeships registered for the project and expressed their willingness to participate in all stages.

According to Junior-Edu-Żywienie (JEŻ) Project theory, grades 1st–6th of primary schools were invited to participate in the quantitative and qualitative research. In total, quantitative research was conducted among 27,295 students, including 14,600 (53.5%) 1st-3rd grade students, aged 7–9 years (subjected to early school education), and among 12,695 (46.5%) adolescents aged 10–12 years at 4th–6th grades. Additionally, the parents of all recruited children were invited to participate in the study, and 17,070 parents responded to the invitation, which constituted 62.5% of the total. At the same time, teachers from schools signed up in the project were also invited to take part in the study. Finally, 2616 teachers from 2218 schools participated in this research. Detailed data divided into macroregions are presented in Table 1. 

Moreover, the food and nutrition curricula were also assessed in the schools participating in the project, including meal organization in school canteens, as well as food assortment available in school shops and vending machines, etc. 

Inclusion and exclusion criteria for students, their parents, and teachers are presented in Table 2. 

### 2.3. Quantitative Research—Methods, Tools, and Data Collection

Quantitative research was conducted among students in two age groups, their parents, teachers, as well as persons responsible for school food environment, e.g., headmasters, intendants, canteen workers, etc. We used the questionnaires developed for the specific purposes of the project but based on validated and published questionnaires [35,36,37]. The questionnaires are listed in Table 3 and described in detail in Section 2.3.2.

#### 2.3.1. Pilot Study

All questionnaires were checked in pilot studies conducted in all study groups, residing in all three environments, i.e., village, town, and city. Researchers noted all questions or ambiguities indicated by respondents. Then, questionnaires were corrected accordingly and approved for the final protocol. 

#### 2.3.2. Questionnaires Description

##### Questionnaires for Children Aged 7–9 Years

Two questionnaires were developed to measure (1) basic nutrition knowledge (Nutrition Knowledge Questionnaire—NKQ) and (2) preferences and the frequency consumption of selected foods (Food Preferences & Food Frequency Questionnaire—FP&FFQ) in children aged 7–9 years. 

The NKQ consisted of 15 questions started mostly with the expression “Do you think that …” and one out of three answers was supposed to be chosen: “Yes”, “No” or “I don’t know”. The themes varied in level of difficulty and included, among others, the dietary guidelines for vegetables and fruit, fish, dairy, and sweets (e.g., “Do you think that vegetables and fruit should be eaten at least 5 times a day?”, “Do you think that sweets can be eaten every day?”); recommendations for physical activity (e.g., “Do you think that you should be physically active every day, e.g., ride a bike, dance, go for walks?”); the relationship between health and food or eating behaviours (e.g., “Do you think nuts, pumpkin and sunflower seeds are good for health?”, “Do you think that eating breakfast before school will give you the power to learn and play?”); and some sustainability related issues (e.g., “Do you think that meat and cold cuts can be replaced with beans, lentils or peas?”). 

The FP&FFQ focused on two questions: child’s liking and consumption frequency of selected foods/food groups. For the first purpose, a 3-point smile face Likert scale was applied with a smiling, neutral, and sad faces meaning “I like”, “I am not sure”, and “I dislike”, respectively. For the second purpose, four levels of frequency consumption were used: everyday, a few times a week, a few times a month, and never/almost never. In total, 28 food items were included, e.g., white bread, whole grain bread, milk, flavoured yogurts, cottage cheese, vegetables, fruit, meat, fish, eggs, sweets, salty snacks, fast food, water, carbonated sugar-sweetened beverages, non-carbonated sugar-sweetened beverages, fruit juices, etc. This questionnaire included two additional questions, namely if the child is interested in nutrition issues (one-choice question with “Yes” or “No” answers) and what are the main sources of his/her nutrition knowledge (multiple choice question with answers like TV, internet, books, school, parents/grandparents, friends). 

Both questionnaires were illustrated with colourful pictures to make them inviting and attractive to children. Language and style as well as letter fonts were adjusted to the perceptive capabilities of early school-age children.

##### Questionnaire for Adolescents Aged 10–12 Years

The questionnaire used in adolescents aged 10–12 years was based on validated tool dedicated to Polish teenagers SF-FFQ4PolishChildren [37], although some elements like nutrition-related knowledge were modified accordingly to current guidelines and recommendations. It consisted of five parts: nutrition-related knowledge, dietary habits, attitudes toward eating, chosen aspects of lifestyle, especially physical activity, screen and sleep time, as well as socio-economic issues.

The first part included 1 open and 20 closed questions related to students’ knowledge on current dietary and physical activity guidelines (e.g., “How much of the healthy plate should be filled by vegetables and fruit?”, “What is the recommended consumption for fish?”, “Recommended daily physical activity, measured by the number of steps, equals …”), relationship between nutrition and health (e.g., “Nuts, e.g., walnuts, almonds and fish improve the functioning of: …”, “Dietary fibre regulates the functioning of:…”), nutritional value of selected food groups (e.g., “A set of products rich in dietary fibre is: …”, “Foods containing healthy fats are:…”), as well as some sustainable issues (e.g., “A diet good for our planet involves limiting the intake of: …”). All closed questions had 4 possible answers to choose, but only one was true, and the fourth was always “I don’t know”. 

The second part focused on dietary habits, mainly on frequency consumption of selected meals (breakfast at home and meal at school) and 12 food groups within the last 12 months prior to survey. The food groups included: dairy products (e.g., milk, yogurt, cottage cheese, cheese), fish (e.g., baked, smoked, fried, canned), fast foods (e.g., chips, pizza, hamburgers), sugar-sweetened beverages (e.g., cola-type, tea-type, water with syrup), fruit or mixed fruit-veggie juices, energy drinks, vegetables (e.g., fresh, boiled, baked, stewed), fruit (fresh or frozen), sweets or confectionery (e.g., cookies, sweets, cake, chocolate bars, chocolate), whole grain products (e.g., bread, oats, brown rice, buckwheat groats), pulses (beans, lentils, chickpeas), and salty snacks (e.g., chips, coated peanuts). The frequency consumption was as follows: never or almost never, less than once a week, once a week, 2–4 times/week, 5–6 times/week, every day, and several times a day. Additionally, students were asked about the impact of the pandemic on their overall eating behaviours, as well as on the consumption of selected food groups in relation to pre-pandemic period. They also assessed their current eating behaviours on 5-point scale.

In the third part, a Three-Factor Eating Questionnaire (TFEQ-13) was presented [38]. The TFEQ-13 consisted of 13 statements within three subscales: Emotional Eating, Uncontrolled Eating, and Cognitive Restraint of Eating. For twelve statements, one out of four answers could be chosen: “definitely yes”, “rather yes”, “rather not”, and “definitely not”. For the last statement (“Indicate on the scale, how much you restrict the food consumption”), an 8-point scale was given with the extreme answers “I don’t restrict at all” and “I always restrict”. 

The next section of this questionnaire included three questions that covered lifestyle issues like student’s physical activity in his/her leisure time (with three answer options: “low”, “moderate”, “vigorous”), his/her screen time (with six answer options from “less than 2 hours/day” till “10 or more hours/day”), and his/her time dedicated usually for sleep (with three answer options: “less than 6 hours/day”, “from 6 to almost 8 hours/day”, “8 or more hours/day”). 

Finally, in the last part, students indicated their gender, school grade, date of birth, sources of their nutrition knowledge, meals eaten together with their family, and the frequency of shared family meals.

##### Questionnaire for Parents

The questionnaire filled out by parents was developed in order to assess his/her child’s eating behaviour and lifestyle as well as parent’s nutrition knowledge and sources of such information. 

It had similar content to the questionnaire dedicated to adolescents aged 10–12 years. It contained the same FFQ (the same food groups and frequencies), questions about the impact of the pandemic on child’s eating behaviours, and three questions on child’s lifestyle. Also, the questions on parent’s nutrition-related knowledge were just as in the questionnaire for adolescents. 

Parents were also asked about the meals eaten together by the family and the frequency of shared family meals. 

Additional issues that occurred only in the questionnaire for parents included the child’s breast-feeding practices—whether the child was breastfed at all, and if so, for how long. They evaluated on a 5-point scale not only their child’s current eating behaviour but their own, too. 

##### Questionnaire for Teachers

The main purposes of the surveys conducted in primary school teachers were to evaluate their nutrition-related knowledge as well as their opinions on students’ eating behaviours and lifestyle and the situation regarding nutrition education at school and the perceived needs in this area. 

In detail, this questionnaire contained the same set of questions regarding nutrition knowledge as the questionnaires for adolescents and parents and additional questions on teacher’s interest in nutrition and evaluation of her/his nutrition-related knowledge on a 3- and 5-point scale, respectively. 

Moreover, it included 8 questions in which teachers could express their opinions on students’ current lifestyle, including nutrition, positive and negative changes that had occurred in students’ eating behaviours in recent years, and the impact of the pandemic on students’ dietary habits. 

Finally, 11 questions were formulated to discover the teacher’s opinion on current state of nutrition education at school, activities undertaken at school in this area, including her/his initiatives, perceived advantages in conducting nutrition education at school, the problems related to nutrition education that exist in her/his school, as well as the possibilities for regular activities regarding nutrition education in primary schools, in general.

The last part contained person-oriented issues, including teacher’s gender, age, tenure, subjects taught at school, field of study completed, additional courses attended to increase her/his teaching competences. 

##### Questionnaire for Evaluation of School Food and Nutrition Environment 

The questionnaire contained 27 closed questions that focused on the food system at school in general and particularly related to meal organization, feeding programs, school shop and/or vending machine, as well as nutrition and physical activity programs conducted at school. 

Meal organization included questions on the number and duration of breaks for breakfast and lunch consumption, the system of school meals preparation (e.g., own kitchen vs. catering) and types of meals offered to students, the number of employees preparing the school meals, the number of students eating the school meals, and the price of particular school meals. 

The part of questionnaire dedicated to the feeding program covered issues like the types of meals and drinks offered, as well as the number of students benefitting from financial support. 

The next questions discussed issues related to school shops, i.e., the food assortment available at school shop, if this assortment was discussed with the school headmaster, whether the school introduced any restrictions or requirements to the assortment of the school shop (according to the Ministry of Health Regulation), and what kind of changes were made. One question also concerned the presence of a vending machine at the school and the available food assortment. 

The later section of the questionnaire focused on nutrition and physical activity programs that were conducted at school in the current school year or that would be planned for the coming school year, specifically if the programs were/would be external or developed by the school. 

The last questions were about the professional trainings/courses that the intendant or cook attended since 2018.

#### 2.3.3. Data Collection 

Data collection among students had an auditory nature and was caried out in the classrooms in the presence of teachers and researchers. Firstly, the general aim of the study was introduced, then questionnaires were distributed to students (only those with parental agreement), and a short instruction on how to fill out the questionnaire was given. In the case of 1st–3rd grade students, questions were read aloud one by one, and children completed the questionnaire by themselves. In the meantime, teachers and researchers checked if each child kept up with their responses. After completion of NKQ, the FP&FFQ was given and explained in a short manner. Children marked their chosen answers individually at their own pace while researchers and teachers were controlling the completion and helped when it was needed. At the end, both questionnaires were checked for completion and coded. It took about 40–45 min, i.e., 1 class hour, to complete both tasks. Students from 4^th^ to 6th grade completed only one questionnaire and completed it on their own. If necessary, detailed explanations were provided, avoiding formulating any suggestive answers. Each questionnaire was coded and verified by researchers if all responses were given. It took about 30–45 min to complete the questionnaire, depending on the grade. 

Parents/caregivers filled out paper versions of questionnaires which had been coded by researchers according to the codes assigned to their children for further child–parent data matching. Questionnaires were completed at home and returned to school within 1–3 weeks of the research conducted at a given school. In each school one teacher was responsible for gathering those questionnaires and sending them to researchers. 

The questionnaire in a paper form was also used to obtain data on the school food environment. It was filled out by the school headmaster or delegated person(s) mostly on the day when research among students was conducted. Researchers could verify answers for some questions at the place, e.g., related to the assortment of school shops or vending machines.

On the other hand, the Computer Assisted Self-Interview (CASI) technique was applied to collect the data among teachers. A questionnaire dedicated to teachers was prepared with Microsoft Forms in advance and a QR code was given to teachers and, additionally, the link was sent via email to all primary schools that had signed up to participate in the Project. 

### 2.4. Qualitative Research 

In order to complement the results obtained in the quantitative research, qualitative research was also conducted among students aged 7–12 years, their parents, and primary school teachers, as well as persons responsible for school food environment, e.g., headmasters, and those preparing meals on school premises or working in catering companies delivering meals to school. 

#### 2.4.1. Focus Group Interviews (FGIs)

Focus Group Interviews (FGIs) were conducted among children aged 7–9 years and adolescents aged 10–12 years, their parents/caregivers, and primary school teachers. They took place in 10 locations throughout Poland. The locations varied in terms of size. The following localities were included in the research: major cities with over 500,000 inhabitants, such as Warsaw, cities with populations ranging from 100,000 to 500,000, including Białystok, Lublin, Kielce, Ostrowiec Świętokrzyski, and Nowy Sącz; small towns with population up to 50,000—Brańszczyk; and villages with population sizes ranging from 330 to 2000, such as Rosko, Czachówek, and Poręba. 

Each focus group consisted of 6 to 8 participants, depending on the school’s capacity and parental agreement for the student’s participation in the study.

The inclusion criteria cover the consent of both the student and his/her parent/caregiver and children with serious health problems or who were on special diets (e.g., elimination ones), as well as parents/caregivers of those children were excluded from the focus groups. 

The research focused on identifying the attitudes of students and parents towards food, nutrition, and physical activity, including the problems and challenges that parents face in terms of feeding their children. The issues referred to questionnaires used in quantitative research among students from both age subgroups and their parents. 

The main themes of FGI with teachers covered, among others, the current state of nutrition education at schools, including opinions on existing programs and students’ nutritional knowledge; the sources of nutrition information and their reliability; the need for the implementation of nutrition education at school; topics, methods, and teaching resources that should be included in nutrition education; and the ways of improving the effectiveness of nutrition education at school and beyond. 

FGI scenarios were developed for the needs of this project and tested in pilot studies in all study groups. FGIs were led by moderators, with one person recording the interviews and taking notes. The main moderator initiated the group discussion by presenting the idea of the discussion and explaining the rules governing the group’s behaviour (confidentiality, respecting each other’s opinions, not interrupting the statements). Participants could resign from participating in FGIs at any stage, without giving reasons, although such situations were not recorded. The duration of each group discussion was approximately 90 min.

A total of 158 students aged 7–12 years and 101 parents of children/adolescents in this age category, as well as 100 teachers, participated in this research.

The qualitative research among these groups of participants was carried out by the professional company Umbrella Agency Marketing Group, as well as by qualified researchers from the Warsaw University of Life Sciences (WULS-SGGW).

#### 2.4.2. In-Depth Interviews (IDIs)

Qualitative research with the technique of individual In-Depth Interview (IDI) was conducted among school representatives responsible for the management and organization of meals, namely headmasters, intendants, or persons from catering companies, depending on the meal system at the school. This technique was applied because of difficulties with organizing FGIs with those people due to their duties realized in different places and times. The methodology was based on adopted standards and in accordance with the assumptions of this type of research [39,40,41].

The following localities were included in this research: cities > 500,000 inhabitants represented by Warsaw (6 schools), Poznan (2 schools); cities with populations ranging from 100,000 to 500,000 such as Toruń, Białystok, Kielce, Lublin, Ostrowiec Świętokrzyski, Tarnobrzeg, and Nowy Sącz; small towns with population up to 50,000 like Brańszczyk, Trzcianka, Muszyna, Ustka; and villages with population size ranging from 330 to 2000, such as Poręba, Nowodwory, Tylicz, and Zielonka Parcele. 

In total, 24 headmasters and 24 representatives responsible for meal organization from 24 schools participated in IDIs. 

The issues raised during IDIs concerned: the organization of school meals, the type of meals available at school and the amount of meals served a day, assessment of customer satisfaction with the meals offered, the scale of food waste, the operation of school shops and vending machines, ideas about the ideal school food system, meals organization during the COVID-19 pandemic, identification of problems in the food and nutrition organization, as well as prospects and needs for changes in this area. Data were collected by WULS’s researchers responsible for conducting the Junior-Edu-Żywienie (JEŻ) Project. 

#### 2.4.3. Qualitative Data Analysis

The group discussions of both types (FGI and IDI) were audio recorded with the consent of the participants, which allowed for subsequent analysis of the participants’ verbatim statements (transcriptions). Transcripts of interviews and notes made during moderation were coded and analysed by researchers.

The material obtained from group discussions was analysed using the principles of grounded theory, with an emphasis on discovering recurring themes that emerged as a result of group interactions. Data analysis was conducted in a seven-step approach: familiarization with the data, thematic coding, identification of sub-themes within the main framework, review and revision of sub-themes, definition and naming of sub-themes, analysis and interpretation of patterns throughout the data area, and combination of sub-themes into dominant contextual domains.

### 2.5. Anthropometric Measurements

Among children and adolescents aged 7–12 years whose parents/caregivers gave consent, anthropometric measurements were conducted. Body weight (BW, kg), height (H, cm), waist circumference (WC, cm), hip circumference (HC, cm), and hand grip strength (HGS, kg) with the precision to the nearest 0.1 kg, 0.1 cm, 0.1 cm, 0.1 cm, and 0.5 kg, respectively, were measured. Measurements were taken twice, and when the differences between two values were greater than 0.5 kg for body weight, 1 cm for height, 1.5 cm for waist or hip circumferences, and 2 kg for hand grip strength, third measurement was taken and the two closest values were chosen. All measurements were in accordance with the International Standards for Anthropometric Assessment [ISAK] recommendations. 

Additionally, body composition was analysed in children and adolescents, except those with arm or leg deformations that prevented proper contact with the electrodes. 

All measurements were taken between 8 am and 12 pm with professional equipment and measuring tape of the same type in all schools. 

Table 4 provides detailed information regarding the anthropometric measurement procedures. 

The distribution of children participating in anthropometric survey, divided into two age subgroups and macroregions, is presented in Table 5. In total, 18,521 students aged 7–12 years were examined, including 9973 aged 7–9 (53.8%) and 8548 aged 10–12 (46.2%). 

### 2.6. Outcome Measures

#### 2.6.1. Dietary Habits 

The dietary habits of children and adolescents were evaluated using food frequency questionnaire filled out by parents. For the comparison, the results achieved from adolescents were similarly analysed. 

The consumption frequency of all food groups was classified as meeting or not meeting the criteria of A Pyramid of Healthy Nutrition and Physical Activity and the dietary guidelines for Polish children and adolescents [48,49]. 

Data from food frequency consumption were converted into real numbers and expressed as daily frequency (times/day) according to the following scoring:Never or almost never: 0 times/day.Less than once a week: 0.06 times/day.Once a week: 0.14 times/day.2–4 times/week: 0.43 times/day.5–6 times/week: 0.79 times/day.Every day: 1 time/day.Several times a day: 2 times/day.

Three diet quality indexes were calculated to comprehensively evaluate the children and adolescents’ diet quality: “pro-Healthy Diet Index” (pHDI).“non-Healthy Diet Index” (nHDI).“Diet-Quality Index” (DQI).

The indexes were established in *A priori* approach on the basis of usual food frequency consumption within last 12 months [35]. The pHDI included six food groups with a potentially beneficial impact on health: dairy products, whole grain products, fish, pulses, vegetables, and fruit. On the contrary, the nHDI included five food groups with a potentially negative impact on health: fast foods, salty snacks, sweets, sugar-sweetened beverages, and energy drinks. Finally, DQI involved eleven food groups, including six ones with a potentially beneficial and five with a potentially negative influence on health.

The pHDI and nHDI were calculated by summing up the daily frequencies of food groups indicated above. In order to standardize the range of both indexes and simplify their interpretation, the results were expressed in % points (in scale from 0 to 100). The overall DQI was calculated as the sum of all weighted pHDI components with a plus sign and all weighted nHDI components with a minus sign. The range of this index varied from −100 to +100 points. 

Two various ideas in categorizing diet quality scores were applied, namely *A priori* and *A posteriori* approaches, both giving three levels of each diet quality. 

In the *A priori,* approach categories such as low, moderate, and high diet adherence were distinguished for pHDI and nHDI, with the ranges of points <33.3%, 33.3–66.7%, and >66.7%, respectively [35]. 

For DQI, the range of points and interpretation were as follows: −100–−26: high intensity of nonhealthy dietary characteristics.−25–26: low intensity of both nonhealthy and healthy dietary characteristics.26–100: high intensity of healthy dietary characteristics.

*A posteriori* approach was based on tertile distributions of all three indexes, with the bottom, middle, and upper tertiles. Those analyses would be conducted when all data were collected and verified. 

Moreover, dietary and lifestyle patterns would be identified among children and adolescents using advanced statistical methods. 

#### 2.6.2. Nutrition-Related Knowledge

The state of nutrition-related knowledge in children, adolescents, their parents, and teachers was also assessed in this Project. Correct answers indicated in questionnaires were scored with 1 point and wrong or “don’t know” answers, as well as missing data, were scored with 0 points. Points were summed up for each respondent and expressed as percentage of correct responses. The maximum number of available points depended on the population group and equalled 15 points for children aged 7–9 years and 20 points for adolescents, parents, and teachers. 

Similar to diet quality interpretations, two approaches for knowledge categorization were used—*A priori* and *A posteriori* approaches, both giving three levels of nutrition-related knowledge. 

The *A priori* approach indicated insufficient, sufficient, and good level of nutrition knowledge with the ranges of points <33.3%, 33.3–66.7%, and >66.7% of correct responses, respectively [35]. 

In the *A posteriori* approach, tertile distribution of data was used and three levels of nutrition knowledge were distinguished: lowest, moderately low, and higher, separately for all subpopulations. 

#### 2.6.3. Nutritional Status

We assessed the nutritional status of children and adolescents based on the anthropometric measurements, body composition, and hand grip strength, according to reference values and interpretations given in Table 6.

Further, body mass index (BMI, kg/m^2^) and waist circumference to height ratio (WHtR) were calculated. First index is a measure of excessive body weight (and general obesity), while the second serves as a measure of central obesity and is useful in predicting cardiovascular risk factors both in children and adults [50].

The BMI and WHtR indices were categorized in accordance with the cut-off values for children and adolescents depending on age and gender based on percentile charts, assuming values between the 10th and 90th percentile as normal [50,51].

Z-scores of WC, BMI, and WHtR as well as HGS were calculated to achieve mean equal 0 and standard deviation (SD) equal 1. Z-scores were categorized as follows: <−1, −1 to 1, and >1 SD.

**Table 6 nutrients-16-00004-t006:** Description of anthropometrics reference values.

Parameter (Units)	Reference Values
Height (H) (cm)	Height results were assessed based on WHO height standards for children, current and representative for the Polish population of children and adolescents aged 3–18 (percentile charts) Values between the 10th and 90th percentile were considered normal, and values between the 3rd and 10th percentile and between the 90th and 97th percentile were considered risky [51]
Body weight (BW) (kg)	Body weight results were assessed based on WHO growth standards for children, current and representative for the Polish population of children and adolescents aged 3–18 years (percentile charts) Values between the 10th and 90th percentile were considered normal, and values between the 3rd and 10th percentile and between the 90th and 97th percentile were considered risky [51]
Waist circumference (WC) (cm)	Waist circumferences were compared with the Polish reference values (percentile charts) [43,50] Reference values for healthy children and adolescents: <90th percentile
Hip circumference (HC) (cm)	Hip circumferences were compared with the Polish reference values (percentile charts) [43,50] Reference values for healthy children and adolescents: <90th percentile
Body composition (BC)	Skeletal muscle mass and fat-free mass were compared with the available reference values [52] Fat mass was compared with the reference values provided in the TANITA manual based on McCarty et al. [44,53]
Hand grip strength (HGS) (kg)	Hand grip strength was compared with the available reference values. Percentile charts depending on age, hand dominance, and gender were used [45] Values between the 10th and 90th percentile were considered normal

### 2.7. Statistical Analysis

Statistical analysis was conducted with Statistica 13.3 software (Tulsa, OK, USA; StatSoft, Krakow, Poland) and a *p*-value of ≤0.05 was considered to indicate statistical significance.

Categorical variables were presented in absolute (*n*) and relative frequencies (%), while continuous variables as means and standard deviations or medians with inter-quartile range for normally or non-normally distributed variables, respectively. 

Data were given for final sample as well as stratified by, e.g., age group and gender, socio-economic status, and macroregions.

The normality of variables distribution was verified with the Shapiro–Wilk test before the statistical analysis. The comparisons among groups were conducted with Student’s *t*-test, Mann–Whitney U-test, one-way ANOVA test, Kruskal–Wallis test, depending on the distribution and number of groups, as well as with chi-square test for categorical data analysis. In addition, the partial correlation between anthropometric indices, dietary indexes, and nutrition knowledge score was investigated with the Spearman correlation test. Further multi-dimensional statistical analysis, like Cluster Analysis, Principal Component Analysis, regression analysis, etc., will be performed to achieve additional outcomes of this study. 

## 3. Nutrition Education Program

Another main goal of the project was to develop and implement a nutrition education program for 1st–6th grade primary school students across Poland. The whole idea of planned educational content, methods, teaching resources, etc., was evidence-based on scientific results obtained in quantitative and qualitative research conducted among children and adolescents, their parents, as well as teachers and other school actors responsible for the school food and nutrition environment. 

The content and tools were dedicated to four target groups, namely children aged 7–9 years, adolescents aged 10–12 years, parents/caregivers, and, mostly, to primary school teachers. 

Materials for teachers, e.g., lesson scenarios, separately for both students’ age subgroups, are designed and their themes correspond to needs and requirements that emerged in the quantitative and qualitative research. Teaching resources including, e.g., presentations, films, rebuses, crosswords, and worksheets are prepared adequately to the scenarios.

Handbooks for teachers and parents/caregivers are developed to increase their nutrition-related knowledge, attitudes, and awareness. It also covers a broad range of topics, starting from dietary guidelines for children and adolescents, the role of a healthy lifestyle and diet in a child’s proper physical, mental, and social development, and ending with the importance of sheared family meals. 

In order to increase teachers’ nutrition-related knowledge, skills, and confidence, they participated in certified trainings including 40 h online and 10 h stationary. 

For children and adolescents, attractive materials are established like comics, cookbooks, and mobile applications (EduApp). In the cookbooks, many recipes of dishes for everyday meals and special occasions are presented. Additionally, it contains new technology options like the free Runvido application. On the other hand, a special mobile application is available for adolescents which contains short videos and quizzes that will improve their knowledge and help in introducing healthy behaviours in their everyday lives. 

Additionally, we will formulate recommendations for school headmasters, intendants, cooks, and other persons working in school canteens and catering companies supplying meals to primary schools to improve the overall school food curriculum.

## 4. Discussion

Presented herein, the Junior-Edu-Żywienie (JEŻ) Project was developed to assess the dietary habits, nutrition-related knowledge and attitudes, and nutritional status of Polish students aged 7–12 years; to evaluate the nutrition-related knowledge of their parents/caregiver and primary school teachers, as well as to assess the school food and nutrition environment. The final goal of this Project was to develop a nutrition education program that will improve children’s nutritional knowledge, dietary habits, nutritional status, and health, including a decline in obesity prevalence, as a further perspective.

Childhood obesity is a global problem, especially in low- and middle-income countries, and is one of the most important public health challenges of the 21st century [14]. The prevalence of excessive body weight among 6-to 9-year-old Polish children is one of the highest in Europe and is the highest in Eastern Europe [54]. Moreover, Poland has the highest ratio of child obesity, with an increase prognosed for the period 2020–2035 (4.8% for each 5 years) much higher than most European countries (e.g., approx. 2.5% for Germany, Greece, Spain, France; approx. 3.0% for United Kingdom, Austria, Finland, and Norway) and twice as high as the USA (2.4%) [23]. 

Obesity in children can dramatically impact their physical health, social and emotional well-being, and self-esteem, resulting in poorer academic achievement and a lower quality of life [14]. Excessive body weight occurring in childhood might persist till adulthood [55]. Obesity is not only a threat per se but is related to almost 40 major comorbidities, including several forms of cancer, hypertension and cardiovascular disease, diabetes, liver and kidney disease, and several other NCDs. So, the enormous costs of this health condition includes not only the treatment of the above-mentioned diseases but also additional costs related to mental health and neurological conditions, endocrine disorders, as well as lower educational attainment, unemployment, long-term disability, and early retirement [14,19,22,23].

The growing problem of childhood obesity can be mitigated if the community focuses on the causes. Combined intervention is needed to improve the diet and physical activity of the school community and in the family environment. Education about healthy eating, making good nutritional choices, and physical activity will ultimately spread to other aspects of children’s lives. Targeting these causes could reduce childhood obesity over time and lead to a healthier society as a whole [56,57,58].

Most of the identified factors contributing to childhood obesity can be viewed as an empirical conglomeration of familial risk factors like parental attitudes toward obesity, eating behaviours, physical activity, screen time, and parenting by grandparents. Family risk factors, as an integrated concept, may be helpful in promoting future family programs for the prevention and early detection of individuals potentially at high risk of obesity in early life. This would enable the provision of flexible and personalized interventions that take into account children’s personal behaviours and needs [56,59].

Therefore, in scientific milieu and at the level of governmental and non-governmental organizations, the need for a multidisciplinary and personalized approach to lifelong lifestyle changes is emphasized to guarantee children the best prospects for the future [60]. World organizations striving to develop strategies to improve population health draw attention to the need to take systemic actions to improve the living environment, which may reduce the epidemic of overweight and obesity [18,61]. 

For that reason, shaping dietary habits in a health-promoting way from an early age becomes extremely important. In light of increasing overweightness and obesity among children and adolescents, it is necessary to undertake health-promoting education activities covering the entire school environment, namely students, teachers, parents, and persons responsible for school meals preparation [3,62].

Numerous studies conducted in Europe and in North America show that school environment has great potential for promoting a healthy lifestyle, including healthy dietary patterns [33,63,64,65]. A higher amount of nutrition education classes within the subject “Nutrition and Household” was significantly associated with a higher intake of wholegrain bread and a lower intake of meat and of energy drinks in Austrian students [66]. Studies also highlight the need for collaboration between the school and family in this matter [63]. A survey conducted in Finland revealed that support from principals and colleagues was the most important facilitator of food education and lack of time was the barrier [64]. Moreover, the commitment of the whole school and principal’s role were underlined to be crucial in the implementation of food education. American teachers perceived nutrition education important on one hand, but on the other they indicated the need for tailored nutrition materials, qualified nutrition personnel, and school stakeholder support to promote nutrition education at schools [33]. The necessity of support and training from experts was also pointed out by Greek teachers [63]. However, the review paper indicated the requirement of conducting more research to inform evidence on the effectiveness of integrative nutrition education for both teacher and student outcomes [67].

Thus, appropriate actions introduced by parents and their involvement in the process, as well as formal and informal nutrition education at school, may be the most effective approach in shaping healthy behaviours among school children. Proper nutrition education for parents and teachers who would receive the useful tools to spread among students the nutrition education in attractive and effective forms is a need. We believe that this will be achieved due to the development and introduction of the Junior-Edu-Żywienie (JEŻ) Project, with its scientific-based background and multicomponent input. The quantitative research conducted on an extensive nationwide scope, and additional data obtained in quantitative research, would enable us to formulate a diagnosis and identify the main problems and, finally, to prepare the appropriately targeted “remedy”.

## 5. Conclusions

Our research may contribute to the identification of subpopulations of children and adolescents at risk of excessive body weight and adiposity, as well as undernutrition and poor muscle structure, and define the predictors related to diet, lifestyle, nutrition knowledge and attitudes, as well as sociodemographic factors that affect body weight and composition disturbances. It will allow us to identify the scale of the problems, including children’s inadequate eating behaviours and insufficient nutrition knowledge among school children, their parents/caregivers, and teachers.

The results also allow us to design and develop a multifaceted educational program based on reliable knowledge and adapted to the perception of target groups. This screening study in a group of children and adolescents from primary schools will fill this gap and develop methodological guidelines for educational and preventive activities. The nutrition-related knowledge and awareness of teachers working in varied environments all over Poland will increase due to attending a broad range of developed training courses. Moreover, permanent educational activities undertaken at school by teachers and at home by parents can provide a good foundation for students’ healthy eating over a longer period. Therefore, the next step needed is to introduce regular and obligatory nutritional classes or subjects dedicated to health and nutrition to the school curriculum.

This study will additionally provide evidence-based support for health care to promote the proper development of the younger population and reduce the risk of diet-related diseases in adulthood in Poland. The results of our study can be implemented as an important public health action.

## Figures and Tables

**Figure 1 nutrients-16-00004-f001:**
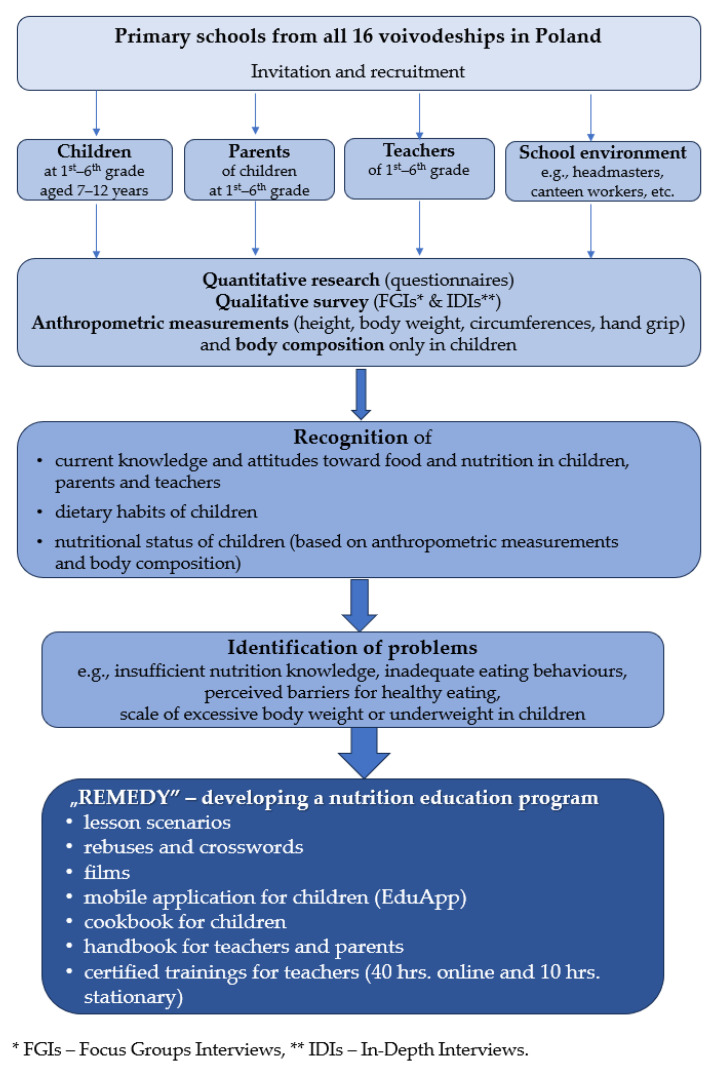
Scheme of stages and activities in Junior-Edu-Żywienie (JEŻ) Project.

**Figure 2 nutrients-16-00004-f002:**
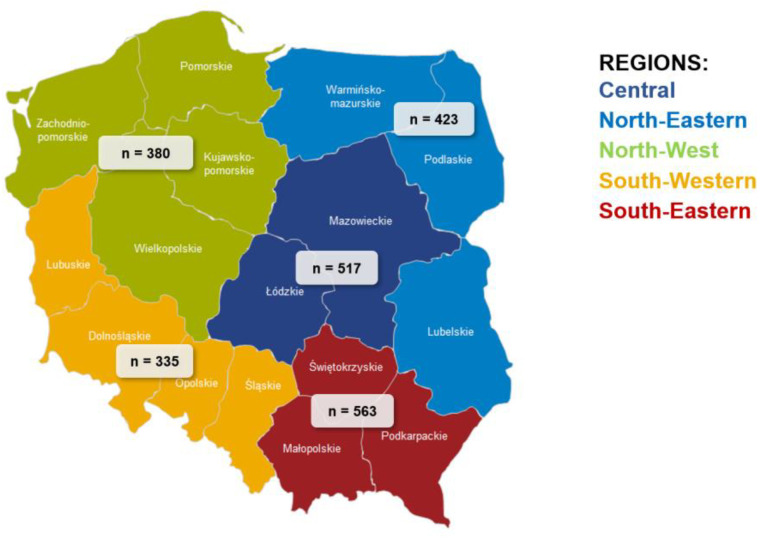
Number of schools participating in the project according to macroregions.

**Table 1 nutrients-16-00004-t001:** Structure of the population groups according to macroregions [n (%)].

Macroregion:	Children/Adolescents		Parents/Caregivers	Teachers
	Aged 7–9 Years	Aged 10–12 Years		
Central	3980 (27.3)	3113 (24.5)	4105 (24.0)	598 (22.9)
South-Eastern	3194 (21.9)	2767 (21.8)	4166 (24.4)	642 (24.5)
South-Western	2017 (13.8)	1548 (12.2)	2167 (12.7)	322 (12.3)
North-Eastern	3481 (23.8)	3243 (25.5)	3738 (21.9)	696 (26.6)
North-West	1928 (13.2)	2024 (16.0)	2894 (17.0)	358 (13.7)
Total	14,600	12,695	17,070	2616

**Table 2 nutrients-16-00004-t002:** Inclusion and exclusion criteria for the study.

Criteria	Children/Adolescents	Parents/Caregivers	Teachers
Inclusion	- students at 1st–6th grades of primary schools - aged 7–12 years - both genders - having the consent of their parents/caregivers - willing to participate in the study - using Polish language	- parents/caregivers of students participating in the study - adults without upper age limit - living with a child - both genders - willing to participate in the study - using Polish language	- persons teaching at 1st–6th grades of primary schools - both genders - adults without upper age limit - willing to participate in the study
Exclusion	- students of other than primary schools - age: <7 or >12 years	- minors - not living with their child or not responsible for child’s feeding	- interns in primary schools - not using Polish language

**Table 3 nutrients-16-00004-t003:** Methods and tools used in quantitative research.

Objectives	Methods and Tools	Population Group		
		Children/ Adolescents	Parents/ Caregivers	Teachers
Assessment of children’s dietary habits	Food frequency method using validated questionnaires:			
	KomPAN^®^ [35]		x	x
	SF-FFQ4PolishChildren [36,37]	x		
	Food Preferences & Food Frequency Questionnaire for children aged 7–9 years	x		
Assessment of children’s lifestyle including physical activity, screen, and sleep time	Questionnaire method:			
	KomPAN^®^ [35]	x	x	
Assessment of knowledge toward food and nutrition in children/adolescents, parents, and teachers	Questionnaire method:			
	KomPAN^®^ [35]	x	x	x
	SF-FFQ4PolishChildren [36,37]	x		
	Nutrition Knowledge Questionnaire for children aged 7–9 years	x		
	Food Preferences & Food Frequency Questionnaire for children aged 7–9 years	x		

**Table 4 nutrients-16-00004-t004:** Description of parameters, measurement methods, and equipment.

Parameter (Units)	Procedure, Accuracy, and Equipment
Height (H) (cm)	Measurement with the head in horizontal Frankfort plane Recorded with a precision of 0.1 cm A portable stadiometer (TANITA Corporation. Tokyo, Japan) Procedure in accordance with adopted recommendations [42]
Body weight (BW) (kg)	Measurement in light indoor clothes without shoes Recorded with a precision of 0.1 kg Electronic digital scale (TANITA Corporation. Tokyo, Japan) Procedure in accordance with adopted recommendations [42]
Waist circumference (WC) (cm)	Measurement at the point midway between the iliac crest and the costal margin (lower rib) on the anterior axillary line in a resting expiratory position Recorded with a precision of 0.1 cm A stretch-resistant tape that provides a constant 100 g tension (SECA 201, Hamburg, Germany) Procedure in accordance with adopted recommendations [43]
Hip circumference (HC) (cm)	Measurement around the widest part of the buttocks, with the tape parallel to the floor Recorded with a precision of 0.1 cm A stretch-resistant tape that provides a constant 100 g tension (SECA 201, Hamburg, Germany) Procedure in accordance with adopted recommendations [43]
Body composition (BC)	Body composition analysis was performed in a standing position. Body weight was evenly distributed over both feet. In light clothes, with no shoes, socks, tights, jewellery, or heavy items in pockets (e.g., phone, wallet) at ambient room temperature The professional TANITA MC-780 S MA multi-frequency, portable, segmented body composition analyser (TANITA Corporation, Tokyo, Japan). The analyser has an European Certificate of approval No EU CE 0122 Individual components of body composition divided into right and left arm and right and left leg and trunk Before each measurement, the electrodes were thoroughly wiped with an appropriate disinfectant Procedure in accordance with TANITA’s instructions [44]
Hand grip strength (HGS) (kg)	Before the measurement, a demonstration of the use of the dynamometer was performed. The dominant hand was identified by asking which hand is used for writing or painting [45] Measurement in the standing position, the arm was allowed to move from 180° of flexion to near 0° with maximal effort [46] Recorded with a precision of 0.5 kg Hydraulic hand dynamometer (SAEHAN Corporation, Asan-si, Republic of Korea, Masan-Korea-type SH 5001) Rest for approximately 2 min between each measurement for each hand was applied to control the effect of fatigue [47] Procedure in accordance with adopted recommendations [45]

**Table 5 nutrients-16-00004-t005:** Number of children participating in anthropometric measurements according to macroregions [n (%)].

Macroregion:	Children/Adolescents		Total
	Aged 7–9 Years	Aged 10–12 Years	
Central	2694 (27.0)	1993 (23.3)	4687 (25.2)
South-Eastern	1660 (16.7)	1447 (16.9)	3107 (16.8)
South-Western	1002 (10.0)	917 (10.8)	1919 (10.4)
North-Eastern	2866 (28.7)	2609 (30.5)	5475 (29.6)
North-West	1751 (17.6)	1582 (18.5)	3333 (18.0)
Total	9973	8548	18,521

## Data Availability

The data presented in this study are available on request from the corresponding author.

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
