# Peer review of "Nutrition-Related Knowledge, Diet Quality, Lifestyle, and Body Composition of 7–12-Years-Old Polish Students: Study Protocol of National Educational Project Junior-Edu-Żywienie (JEŻ)"

_nutrients, 2023, doi:10.3390/nu16010004_

Round 1

Reviewer 1 Report

Comments and Suggestions for Authors

In this manuscript, the authors reported the study protocol of a National Educational Project, Junior-Edu-Żywienie (JEŻ). The content of the manuscript may be interesting, helpful, and informative to the readers. However, the following issues in the manuscript should be addressed.

1. P1, Lines 27-28: Measurements of body weight, height, waist and hip circumferences and handgrip strength were performed in accordance with the ISAK recommendations.

The full name of “ISAK” should be displayed when it is appears for the first time in the manuscript.

2. P1, Line 37: It is inappropriate and unnecessary to take “hand grip strength” as a keyword.

3. P3, Lines 121-122: The consent form was distributed among parents/caregivers during the school meetings. Participants who refused to participate in the study attended other school activities.

The second sentence is unclear and redundant.

4. P3, Lines 125-126: Within the Junior-Edu-Żywienie (JEŻ) Project a prospective, cross-sectional was carried out in the period from June, 2022 to November, 2023.

It would be better to delete the word “prospective”. The JEZ is a cross-sectional study and is not prospective study in nature.  

5. P3, Lines 126-129: The main assumption was to create a representative sample including 1st-6th grade primary students (aged 7-12 years) from schools located in all 16 voivodeships across Poland, taking into account three environments (villages, towns, and cities) and five macroregions……

The word “assumption” is inappropriate.

6. P4, Lines 147-148: Figure 2 presents the overview of main stages and activities undertaken during Junior-Edu-Żywienie (JEŻ) Project and corresponding to project objectives.

This sentence is redundant.

7. P5, Line 152: Figure 2. Scheme of stages and activities in Junior-Edu-Żywienie (JEŻ) Project.

This diagram is unclear. For examples: the description of quantitative and qualitative studies and anthropometric measurements is too simplistic and unclear. The word “DIAGNOSIS” is inappropriate.

8. P9, Lines 289-292: Next 8 questions focused on evaluation of current eating students’ behaviors and lifestyle, positive and negative changes that had occurred in students’ eating habits during last years, as well as the impact of the pandemic on students’ eating behaviors and lifestyle.

“Current eating students’ behaviors and lifestyle”?

9. P10, Lines 324-325: Last questions were about the trainings that intendant or cook attended during the last period: whether they participated in any courses at all and if so, what kind they were.

What is the meaning of “during the last period”?

10. P14, Lines 464-466: Categorical variables were presented in absolute (n) and relative frequencies (%), while continuous variables as means with 95% confidence interval (95% CI) or medians with inter-quartile range for normally or non-normally distributed variables, respectively.

Normally distributed continuous variables should be described in terms of mean (standard deviation).

11. P14, Lines 470-477: The comparisons among groups were conducted with Student’s t-test, Mann-Whitney U-test, one-way ANOVA test, Kruskal-Wallis test, depending on the distribution and number of groups, as well as with chi-square test for categorical data analysis. In addition, the partial correlation between anthropometric indices, dietary indexes, and nutrition knowledge score was investigated with the Spearman correlation test. Further multi-dimensional statistical analysis, like Cluster Analysis, Principal Component Analysis, regression analysis, etc. will be performed to achieve additional outcomes of this study.

The introduction of the statistical analysis methods should be more specific.

12. “3. Outcome Measures” should be a part of “2. Materials and Methods”.

13. It is suggested that the authors describe the socio-demographic characteristics of students, parents, teachers, etc., who participated in the JEÅ», and compare their characteristics with those of the individuals who did not participate in, especially who refused to participate in, the JEÅ» to assess the representativeness of the JEÅ» participants. 

Comments on the Quality of English Language

The language of the manuscript should be further improved.

Author Response

Dear Reviewer,

Thank you very much for all your valuable comments and suggestions. We have tried to address them accordingly in the manuscript and marked them in green. We hope the changes incorporated into the text and our responses will be satisfactory.

Kind regards

Authors

Reviewer 2 Report

Comments and Suggestions for Authors

The present manuscript of a study protocol is focused on 3 main goals:

1/Assessing the dietary habits, nutrition-related knowledge and attitudes, and nutritional status of Polish students aged 7-12 years;

2/Recognizing nutrition-related knowledge among their parents and teachers as well as the school food and nutrition environment;

3/Developing of a nutrition education program.

The protocol is very detailed and adequately prepared. It makes really good impression the scale of the research which covers primary schools in the entire country, including rural, small-town, and metropolitan areas. There are only minor details to be clarified or fixed:

Lines129-135 and fig.1 Authors needs to clarify whether this division of the included regions is according to NUTS or based on other principle

Fig 2 is too small

It will be good to present he English version of each type of questionnaire (for kids, for parents, for teachers and school headmasters) as supplement at the end of the paper so that it can be used by other specialists in the field.

Line 501 last 12 months

Author Response

Dear Reviewer,

Thank you very much for all your valuable comments and suggestions. We have tried to address them accordingly in the manuscript and marked them in yellow. We hope the changes incorporated into the text and our responses will be satisfactory.

Kind regards

Authors

Reviewer 3 Report

Comments and Suggestions for Authors

1. I recommended diminishing the number of keywords

2. The study started in April 2022 and the authors declared that the study started in June 2023. They have to clarify that. Because they said that the pilot study was performed in April-May 2022

3. Table 1and 5, should better indicate N and %

4. Table 6 maybe must be in Material and methods

5. The results must be improved, with a better explanation of the obtained results

Author Response

Dear Reviewer,

Thank you very much for all your valuable comments and suggestions. We have tried to address them accordingly in the manuscript and marked them in purple colour. We hope the changes incorporated into the text and our responses will be satisfactory.

Kind regards

Authors
